# Effects of Neuromuscular Electrical Stimulation on Spasticity and Walking Performance among Individuals with Chronic Stroke: A Pilot Randomized Clinical Trial

**DOI:** 10.3390/healthcare11243137

**Published:** 2023-12-11

**Authors:** Sattam M. Almutairi, Mohamed E. Khalil, Nadiah Almutairi, Saud M. Alsaadoon, Dalal S. Alharbi, Sultan D. Al Assadi, Salem F. Alghamdi, Sahar N. Albattah, Aqeel M. Alenazi

**Affiliations:** 1Department of Physical Therapy, College of Medical Rehabilitation, Qassim University, Buraydah 52571, Saudi Arabia; freephysio77@gmail.com (M.E.K.); sahr.nb@gmail.com (S.N.A.); 2King Fahad Specialist Hospital, Buraydah 52366, Saudi Arabia; nalmutairi74@moh.gov.sa; 3Department of Rehabilitation Services and Programs, Sultan Bin Abdulaziz Humanitarian City, Riyadh 13571, Saudi Arabia; alsaadoonsa@mngha.med.sa (S.M.A.); daalharbi@sbahc.org.sa (D.S.A.); salassadi@moh.gov.sa (S.D.A.A.); saalghamdi@sbahc.org.sa (S.F.A.); 4Department of Health and Rehabilitation Sciences, College of Applied Medical Sciences, Prince Sattam Bin Abdulaziz University, Al-Kharj 11942, Saudi Arabia; aqeel.alanazi@psau.edu.sa

**Keywords:** cerebrovascular accident, electrotherapy, spastic, stroke

## Abstract

Background: Stroke and its associated complications are a major cause of long-term disability worldwide, with spasticity being a common and severe issue. Physical therapy, involving stretching exercises and electrical stimulation, is crucial for managing spasticity. Therefore, this study aimed to evaluate the effects of neuromuscular electrical stimulation (NMES) combined with a conventional rehabilitation program (CRP) on plantarflexor muscle spasticity and walking performance among individuals with chronic stroke. Methods: A pilot randomized clinical trial (RCT) with two groups (active NMES and placebo) was conducted at the physical therapy departments of King Fahad Specialist Hospital, Buraydah, and Sultan Bin Abdulaziz Humanitarian City, Riyadh, Saudi Arabia (November 2020). The assessor and participants were blinded for the group assignment. The active NMES group received exercise and stimulation at the dorsiflexor muscles on the paretic leg for 30 min for 12 sessions. The placebo group received exercise and sham stimulation at the same position and duration as the active group. Of interest were the outcomes for plantarflexor muscle spasticity measured by the modified Ashworth scale (MAS), gait speed measured by 10 m walk test (10-MWT), and functional mobility measured by functional ambulatory category (FAC). Results: Nineteen participants were randomized into active NMES (*n =* 10) and sham NMES (*n* = 9) groups, with no significant baseline differences. Within the active NMES group, significant improvements were observed in MAS (*p =* 0.008), 10-MWT (*p =* 0.028), and FAC (*p =* 0.046), while only 10-MWT time improved significantly in the sham NMES group (*p =* 0.011). Between-group analysis showed that only MAS was significantly lower in the active NMES group (*p =* 0.006). Percent change analysis indicated a significantly higher increase in percent change for MAS in the active NMES group compared to the sham NMES group (*p =* 0.035), with no significant differences in other outcome measures. Conclusions: This study showed that NMES in the active group led to significant improvements in spasticity, walking performance, and functional ambulation. Further research is needed to determine the ideal parameters, protocols, and patient selection criteria for NMES interventions in stroke rehabilitation.

## 1. Introduction

Stroke and its associated complications are a leading cause of long-term disability worldwide, with millions of individuals affected each year [1]. The annual incidence rate of stroke in Saudi Arabia exhibits prevalence with a reported incidence of 29 strokes per 100,000 individuals annually [2]. Stroke is associated with a wide variety of disabling complications, among which spasticity is one of the most common and severe [3]. It often occurs as a result of damage to the central nervous system (CNS) following a stroke [3]. Understanding the underlying mechanisms of spasticity in stroke patients is crucial for optimizing rehabilitation strategies and improving patient outcomes [4].

The primary lesion in the CNS, particularly following a stroke in the cortex, disrupts the normal balance between inhibitory and excitatory pathways, leading to an exaggerated stretch reflex with altered transmission in a variety of spinal cord pathways, which often induces spasticity [3,4]. Spasticity in patients with stroke presents with a range of clinical manifestations that can significantly impact functional abilities and quality of life [5]. The severity of spasticity varies among individuals, ranging from mild muscle stiffness to severe muscle contractures. These physical impairments can lead to limitations in activities of daily living, decreased mobility, and compromised functional independence [6]. 

Current management strategies aim to reduce spasticity, improve motor function, and enhance overall quality of life [7]. Pharmacological interventions, including muscle relaxants and botulinum toxin injections, can provide temporary relief; however, they often have limited efficacy and may be associated with adverse effects [8]. Physical therapy, including stretching exercises, assistive devices, and electrical stimulation, plays a crucial role in managing spasticity, restoring range of motion, and preventing contractures with less adverse effects [9,10]. Neuromuscular electrical stimulation (NMES) can be utilized to reduce spasticity by modulating spinal reflexes, thereby influencing their excitability. Moreover, NMES has the potential to induce neuroplasticity within spinal cord pathways, resulting in adaptive changes that contribute to spasticity reduction. This is achieved through the promotion of reorganization in neural connections facilitated by NMES [4]. A previous study found that the group of patients with stroke who received a combination of sit-to-stand training and transcutaneous electrical stimulation (TENS) demonstrated a significant reduction in ankle spasticity compared to the sham group [11]. This suggests that utilizing TENS along with sit-to-stand training can effectively alleviate spasticity in stroke patients. A systematic review was conducted, analyzing 29 randomized clinical trials, with the aim of evaluating the impact of NMES on spasticity in individuals with stroke. The review revealed that, among the included trials, NMES demonstrated a significant reduction in spasticity in 14 randomized clinical trials when compared to control groups [12]. In a randomized, single-blind, controlled study, the use of NMES resulted in significant reductions in plantarflexor spasticity and improvements in ankle function among subacute stroke patients when compared to the control group. However, no significant improvement was observed in walking time between the groups [13]. Despite these interventions, many patients continue to experience residual spasticity, highlighting the need for further research and innovative treatment approaches. Nevertheless, there have been conflicting studies that have failed to demonstrate improvement in spasticity levels following the use of electrical stimulation [14,15]. Therefore, this study aimed to contribute to the growing body of evidence and facilitate the development of NMES to evaluate the effects of NMES combined with a conventional rehabilitation program (CRP) on plantarflexor muscle spasticity and walking performance among individuals with chronic stroke. 

## 2. Materials and Methods

The study followed the Standard Protocol Items Recommendations for Interventional Trials (SPIRIT) guidelines and checklists in developing the protocol. The research protocol was registered in the ClinicalTrials database with the registration code NCT0467304 (November 2020).

### 2.1. Study Design 

The study design consisted of a randomized clinical trial (RCT) with two groups (active NMES and placebo) and two time points (baseline and post-intervention). The participants were assigned randomly in a 1:1 ratio to either the experimental group or the placebo group. Both the outcomes assessor and participants remained unaware of the group allocation, ensuring blinding.

### 2.2. Study Site

This study was conducted at the physical therapy departments, King Fahad Specialist Hospital, Buraydah, Saudi Arabia, and Sultan Bin Abdulaziz Humanitarian City, Riyadh, Saudi Arabia. 

### 2.3. Participants

As a convenient sample, participants with chronic stroke were enrolled from the outpatient department at King Fahad Specialist Hospital and inpatient at Sultan Bin Abdulaziz Humanitarian City. The study protocol received approval from the national bioethics committee review boards of the Ministry of Health, Saudi Arabia (1442-551803), and the research center of Sultan Bin Abdulaziz Humanitarian City (74-2022-IRB). Following recruitment, an initial screening process was carried out to assess if potential participants met the inclusion criteria. Prior to their participation, informed consent was obtained from all participants. The study was conducted in accordance with the principles outlined in the Declaration of Helsinki. Participants who met the following criteria were included in the study: they must have experienced their first brain stroke between the ages of 18 and 65, with a minimum of 6 months having passed since the stroke to ensure the exclusion of spontaneous recovery effects. Additionally, they should exhibit plantarflexor spasticity on the affected limb with a score of 1 or higher on the modified Ashworth scale (MAS). Lastly, their functional ambulation level should be rated at 2 or higher on the functional ambulation categories scale (FAC) [16]. Participants who did not meet the inclusion criteria were excluded from this study. Additionally excluded were participants with skin problems on the area where the NMES would be applied, significant cognitive impairments that prevented them from following instructions, unstable medical conditions, previous disorders that impacting walking ability, multiple strokes, contraindications to NMES, having received injections of spasticity-reducing medications like Botulinum-A Toxin, being pregnant, or having previously undergone lower limb treatment with functional electrical stimulation (FES) or NMES. Figure 1 presents a visual representation following the CONSORT guidelines for the participant enrollment process in this study.

### 2.4. Data Collection and Procedure

Participants were randomly assigned to either the NMES active or NMES sham group in a 1:1 ratio using an online randomization website (https://www.graphpad.com/quickcalcs/randomize1.cfm, accessed on 2 October 2020). The randomization procedure was carried out by a research assistant who was not part of the intervention or data collection process. Randomized allocations were concealed in sealed envelopes, devoid of any identifying information. Assessors and participants remained blinded to the group allocation. Following the initial evaluation, a research assistant chose an envelope and notified a trained therapist, who had no prior involvement in the study, regarding the allocated group for the respective participant. To maintain blinding, the assessor was banned from attending interventional sessions for both groups, and schedules were carefully managed to minimize contact between participants from both groups.

The evaluation of the plantarflexor muscles spasticity of the involved leg was performed through the MAS [17]. Participants were positioned in a supine, and the assessor assessed the spasticity of the plantarflexor muscles. Starting from the point of maximum ankle plantarflexion, the assessor passively moved within 1 s the ankle joint to the maximum dorsiflexion. In addition, the 10 m walk test (10-MWT) was used to measure preferred walking speed. Participants were asked to walk a 10 m designated distance, with specific zones for acceleration and deceleration, usually 6 m in the middle. The speed calculation focused on the 6 m distance between these zones. The test was completed three times, and the recorded speeds were averaged for analysis. The 10-MWT has demonstrated high reliability and 0.14 m/s was reported as the minimum clinically important difference (MCID) for meaningful change [18,19]. Furthermore, the FAC scale was used to measure the ambulatory function of individuals with neurological conditions [16]. It consists of six levels, ranging from nonfunctional ambulation to normal ambulation. The FAC assesses a person’s ability to walk independently, considering factors such as the need for assistance, walking aids, negotiating stairs, and adapting to different walking conditions. The desired outcomes were evaluated prior to and following a 4-week intervention period, with a 3-day window for assessment.

### 2.5. Intervention

After obtaining informed consent, participants completed a data form providing demographic data, past medical history, past surgical history, and activity level. They were then screened for inclusion/exclusion criteria. Participants who met the criteria were enrolled in the study and evaluated on the outcomes before and after the intervention. Randomization was conducted using the previously mentioned method for allocation sequence generation, resulting in two groups.

Every participant underwent a conventional rehabilitation program (CRP) that included warm-up exercises, muscle-strengthening exercises, stretching exercises, static and dynamic balance exercises, and gait exercises (Table 1). This program was conducted for 45 min per day, three times a week, for 4 weeks. In addition to the CRP, the NMES active group received 30 min of active NMES, while the placebo group received 30 min of sham NMES. Reports on the duration of NMES varied in the literature, but 30 min was deemed acceptable and feasible in a clinical setting to ensure maximum patient adherence.

For active and placebo electrical stimulation therapy, the Gymna equipment (Pasweg 6a, 3740 Bilzen, Belgium) was utilized. The selected frequency of 80 Hz aimed to stimulate dorsiflexor muscle strength without posing a risk of excessive frequency. Table 2 provides the parameters used for active electrical stimulation. NMES was administered through two surface electrodes (6 × 8 cm) placed on saline-soaked sponges. The stimulation intensity was adjusted according to the subject’s tolerance level, aiming to produce visible muscle contractions without causing discomfort. The cathode electrode was positioned over the common peroneal nerve at the fibular head, while the anode was positioned on the ankle dorsiflexor muscle belly, which is located one-third of the distance between the fibular head and the medial malleolus.

Participants remained supine during stimulation. For the placebo group, the placement of the electrodes was the same as for the active group. However, the current intensity was gradually reduced to nothing after a few seconds. Thus, participants initially experienced the sensation of current, but no current was delivered for the rest of the stimulation period. Participants were notified that the stimulation would not be strong and would not have a noticeable sensory response. After each session, participants were checked for any pain or skin irritation beneath the electrodes. Any adverse events were recorded, discussed with the participant, and addressed accordingly. 

### 2.6. Statistical Analysis 

Data were expressed as means and standard deviation for continuous data. For categorical data, counts and percentages were used in the analysis. The chi-square or Fisher exact test was used to compare baseline categorical variables. Normality analysis was conducted to determine the appropriate analysis using the Shapiro–Wilk test due to the small sample size in the current study. The results of the Shapiro–Wilk test show significant results indicating all the outcome measures were not normally distributed at all levels, including pre- and post-test. Therefore, nonparametric tests were selected for analysis. 

For baseline between-group differences, the Mann–Whitney test was used to compare between the active NMES group and the sham NMES group. To compare within-group results, the Wilcoxon signed-rank test was utilized to compare baseline results with post-intervention results for the outcome measures for each group. The Mann–Whitney test was used to compare post-intervention results between the active NMES group and the sham NMES group. To compare the changes between the active NMES group and the sham NMES group, percent changes were calculated for each outcome measure using this formula: 100 × [baseline measures − post-intervention measures]/baseline measures. All analyses were conducted using SPSS version 25 for Mac (IBM Corp., Armonk, NY, USA) with an alpha level of less than 0.05. 

## 3. Results 

A total of 19 participants were included in the current study and randomized into two groups: the active NMES group (*n =* 10) and the sham NMES group (*n =* 9). Table 3 shows the comparison between the active NMES group and the sham NMES group at baseline. The results show that there were no significant differences between groups using the Mann–Whitney test and the Fisher exact test for continuous and categorical data, respectively. 

Table 4 shows the within-group differences for outcome measures using the Wilcoxcon signed-rank test. For within-group differences for the active NMES group, there were significant decreases in MAS score from 2.80 to 1.80 (*p =* 0.008), 10 MWT from 24.2 s to 19.9 s (*p =* 0.028), and a significant increase in FAC from 3.8 to 4.2 (*p =* 0.04). For within-group differences for the sham NMES group, only 10 MWT showed a significant decrease in time from 16.9 s to 15.20 s (*p =* 0.011). 

Table 5 shows the between-group differences for the post-intervention of the outcome measures using the Mann–Whitney test. Only MAS was significantly lower in the active NMES group compared to the sham NMES group (*p =* 0.006). Other outcomes were not significantly different between the active NMES group and the sham NMES group. 

To decrease the effect of baseline differences, a comparison of percent change was analyzed using the Mann–Whitney test. Table 6 shows the results of the between-group differences for percent change in the outcome measures. Only MAS has a significantly increased percent change in the active NMES group (35%) as compared with the sham NMES group (−10.74%), (*p =* 0.035). All other outcome measures showed no significant differences in percent change between the active NMES group and the sham NMES group. 

## 4. Discussion 

This study aimed to evaluate the effects of NMES combined with CRP on plantarflexor muscle spasticity and walking performance among individuals with chronic stroke. The results of this study showed that the active NMES group exhibited significant improvements in plantarflexor muscles spasticity, gait speed, and mobility function. In the sham NMES group, only the gait speed showed a significant decrease from 16.9 to 15.20 s. Concerning between-group differences for the post-intervention outcome measures, the results showed that only plantarflexor muscles spasticity was significantly improved in the active NMES group compared to the sham NMES group. To account for baseline differences, percent change comparisons indicated that the plantarflexor muscle spasticity had a significantly higher percent change in the active NMES group (35%) compared to the sham NMES group (−10.74%). 

The utilization of NMES in the dorsiflexor muscles of the lower limb affected by hemiplegia due to stroke has shown promising results. Similar to this study, in a four-week study, the application of NMES to dorsiflexor muscles resulted in significant improvements in the functional independence-measure motor subscore [20]. This improvement in the motor subscore indicates a notable clinical enhancement in the hemiplegic lower limb following a stroke. Furthermore, NMES reduced static and dynamic spasticity of the plantarflexors, and improved plantarflexion push-off [13,21]. These findings suggest that NMES can effectively address spasticity-related issues and enhance the functional capabilities of the plantarflexor muscles in individuals recovering from stroke, although the specific mechanism responsible for this improvement is not yet fully understood [22]. Nevertheless, numerous studies have indicated that the therapeutic impact of NMES on central nervous system (CNS) lesions could be attributed to central neuroplasticity [23,24,25]. However, these studies have primarily concentrated on investigating supraspinal mechanisms. In the case of stroke patients, who may experience spasticity due to spinal reflex abnormalities and excitability affecting the function of spinal neurons [3,4], there has been a lack of direct research exploring whether the reduction in spasticity induced by NMES is associated with neuroplasticity effects within specific pathways of the spinal cord. It is worth noting that several studies showed no significant change in spasticity [13,21,26,27]. However, it has been argued that variations in the parameters used for electrical stimulation could account for the discrepancies in the results [28]. These parameters include the frequency, intensity, and duration of the electrical stimulation. Additionally, other factors, such as the location of stimulation and individual patient characteristics, may also contribute to the inconsistent findings across studies. 

When comparing our study to the studies conducted by Wang et al. [13] and Yang et al. [21], there were notable differences in the intervention parameters. Specifically, our study utilized a higher frequency (80 Hz), a duration of 30 min, and a sinusoidal waveform, whereas Wang et al. and Yang et al. employed 50 Hz on either the dorsiflexors or plantarflexors. Despite these variations, our study yielded similar positive results, including a reduction in spasticity of the plantarflexor muscles, improved gait speed, and enhanced mobility function. These findings suggest that the use of a higher frequency, 80 Hz, combined with 30 min duration and a sinusoidal waveform, proved effective in achieving the desired outcomes in our study.

Typically, in neurological conditions, frequencies ranging from 20 to 100 Hz are employed to enhance the activation of type I (20–35 Hz) and type II (30–70 Hz) muscle fibers. High frequencies beyond 100 Hz induce rapid muscle fatigue in individuals with post-stroke paralysis [29,30]. The symmetrical biphasic rectangular waveform is commonly used, although some studies have explored square or pulse waveforms [31]. The pulse duration for electrical stimulation typically ranges from 200 to 450 μs (0.2–0.45 ms); commonly, a 350–450 μs range is utilized to activate weakened muscles. Lower pulse durations may provide greater patient comfort but are less effective, while pulse durations exceeding 450 μs may boost the electrical charge to the muscle, depending on the application time. Nevertheless, pulse durations beyond 450 μs may enhance the activation of larger afferent nerve fibers, potentially leading to increased recruitment of central motor neurons [29,30].

These findings contribute to the understanding of NMES as a potential treatment approach for managing spasticity in stroke patients. The lack of significant between-group differences in most outcome measures raises questions regarding the superiority of active NMES over sham NMES in terms of functional improvements. Particularly, several studies report that NMES enhance or reduce spasticity but not functional activity [13,32]. It is possible that the sample size was not sufficient to detect significant differences or that other factors influenced the results, such as variations in stroke severity or individual responsiveness to NMES. Additionally, the absence of significant improvements in walking time between the two groups raises further considerations regarding the impact of NMES on gait performance in stroke patients.

This study has some limitations that need to be considered. The sample size was relatively small, which might affect the interpretation of the results. The lack of follow-up is another limitation that limits the ability to examine the maintained improvements through time. The study was not multicenter, which limited the sample size and potential for generalizability. However, a single-center approach provided better control over variables and minimized potential confounding factors, thus strengthening the internal validity of the study. To overcome these limitations, future research should consider a larger sample size and longer follow-up time. 

## 5. Conclusions

This study demonstrated that NMES, as applied in the active NMES group, resulted in significant improvements in plantarflexor spasticity, walking ability, and functional ambulation. However, only plantarflexor spasticity showed a significant improvement during the comparison between groups. These findings should be interpreted in the context of the existing literature, which has shown mixed results regarding the effectiveness of electrical stimulation in managing spasticity. Conflicting studies have reported varying outcomes, suggesting the need for further research to elucidate the optimal parameters, protocols, and patient selection criteria for NMES interventions in stroke rehabilitation.

## Figures and Tables

**Figure 1 healthcare-11-03137-f001:**
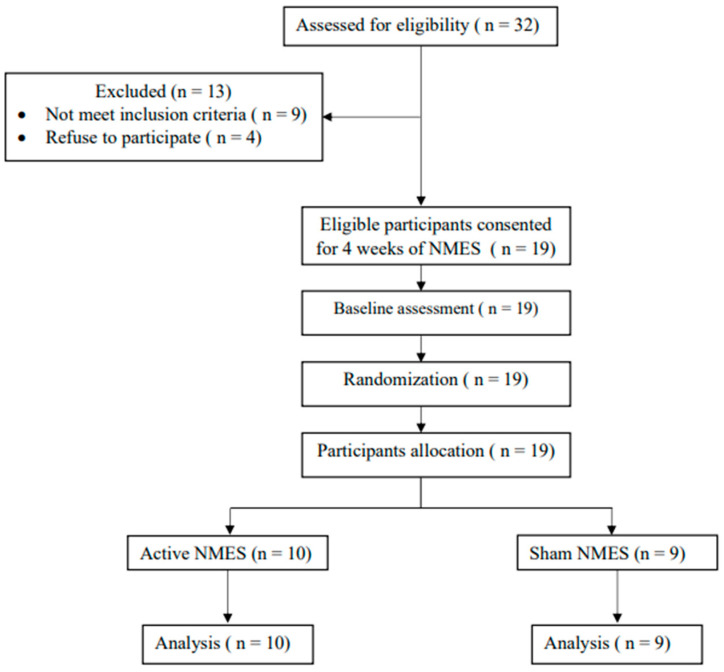
Visual representation following the CONSORT guidelines for the participant enrollment process in this study.

**Table 1 healthcare-11-03137-t001:** Description of the components included in the conventional rehabilitation program.

Components	Time	Description
Warming up	5 min	Bicycling using a stationary bicycle or ergometer.
Stretching exercise	5 min	Unilateral for the following muscle groups: wrist flexors, biceps, pectoral major, shoulder extensors, quadriceps, hamstrings, gastrocnemius, and thigh adductors.
Strengthening exercise	5 min	Upper extremity strengthening exercise using small pulley weight. Lower extremity strengthening exercise using quadriceps chair.
Postural control and balance	3 min	Sit to stand transition with symmetrical weight bearing and trunk rotation.
	3 min	Dynamic balance activity includes low frequency sway and increase weight shifting on the affected side.
	4 min	Gentle perturbations to displace center of gravity (COG) using a gymnastic ball or equilibrium.
Upper extremity control	5 min	Moving the upper extremity with emphasis on scapular motion. For example, hand to mouth, hand to opposite side, and hand functions.
		Grasping and releasing objective.
Lower extremity control	3 min	Pre-gait mat activity includes bridging, hook lying, and lower trunk rotation.
Gait training	12 min	Gait training using parallel bar; gait training includes forward, backward, sideward step, and in crossed pattern.

**Table 2 healthcare-11-03137-t002:** Parameters for active electrical stimulation.

Carrier Wave	2.5 kHz
Burst	80 Hz
On time	5 s
Off time	15 s
Treatment time	30 min
Current type	Constant current
Waveform	Sinusoidal

**Table 3 healthcare-11-03137-t003:** Demographics and baseline clinical measures.

	Active NMES (*n =* 10)	Sham NMES (*n =* 9)	*p*-Value
Age (years), mean ± SD	57.9 ± 9.9	54.7 ± 12.3	0.55
Sex, males, *n* (%)	10 (100)	7 (77)	0.21
Time since stroke (months), mean ± SD	32.5 ± 34	26.44 ± 30	0.69
Side of paralysis, left, *n* (%)	6 (60.0)	7 (77.8)	0.62
MAS, mean ± SD	2.8 ± 0.9	3.4 ± 1	0.16
10-MWT (s), mean ± SD	24.22 ± 21	16.97 ± 7.8	0.84
FAC, mean ± SD	3.80 ± 0.63	3.33 ± 0.86	0.35

Note: NMES—neuromuscular electrical stimulation; SD—standard deviation; %—percentage; *n*—numbers; s—seconds.

**Table 4 healthcare-11-03137-t004:** Within-group differences for outcome measures using the Wilcoxon signed-rank test.

	Active NMES (*n =* 10)	*p*-Value	Sham NMES (*n =* 9)	*p*-Value
MAS, mean difference ± SD	1.00 ± 0.66	0.008	0.22 ± 1.1	0.57
10-MWT (s), mean difference ± SD	4.27 ± 6.7	0.028	1.76 ± 1.42	0.011
FAC, mean difference ± SD	−0.40 ± 0.51	0.046	−0.22 ± 0.66	0.32

Note: NMES—neuromuscular electrical stimulation; SD—standard deviation; *n*—numbers; s—seconds.

**Table 5 healthcare-11-03137-t005:** Between-group differences for the post-intervention of the outcome measures using the Mann–Whitney test.

	Active NMES (*n =* 10)	Sham NMES (*n =* 9)	*p*-Value
MAS, mean ± SD	1.80 ± 0.91	3.22 ± 0.83	0.006
10-MWT (s), mean ± SD	19.95 ± 16.22	15.20 ± 7.15	0.96
FAC, mean ± SD	4.20 ± 0.42	3.55 ± 0.72	0.053

Note: NMES—neuromuscular electrical stimulation; SD—standard deviation; *n*—numbers; s—seconds.

**Table 6 healthcare-11-03137-t006:** Between-group differences for percent change in the outcome measures using the Mann–Whitney test.

	Active NMES (*n =* 10)	Sham NMES (*n =* 9)	*p*-Value
MAS, percent change	35.00	−10.74	0.035
10-MWT (s), percent change	15.59	9.82	0.35
FAC, percent change	−11.11	−12.50	0.78

Note: NMES—neuromuscular electrical stimulation; *n*—numbers; s—seconds.

## Data Availability

Datasets analysis will be available from the corresponding author upon reasonable request after publication of the trial findings.

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
