# Peer review of "Effects of Neuromuscular Electrical Stimulation on Spasticity and Walking Performance among Individuals with Chronic Stroke: A Pilot Randomized Clinical Trial"

_healthcare, 2023, doi:10.3390/healthcare11243137_

Round 1

Reviewer 1 Report

Comments and Suggestions for Authors

The cause is not a stroke but the complications.

Spasticity cannot be called a problem, it is rather a limitation of full activity.

Exercises and electrostimulation may not be crucial.

Walking efficiency was not measured, but rather quality (subjective assessment) or quantity (objective assessment, e.g. 3D gait analysis)

There is no information about who conducted the MAS study or what the group selection process was (random, random, or coin tossing). There is no information about the state of spasticity, how many years after the stroke, how long the exercises lasted and the type of exercises - standard or neurodevelopmental

Comments on the Quality of English Language

Mainly comments after taking into account suggestions in the content

Author Response

Response to Reviewer 1

Comments 1: The cause is not a stroke but the complications.

Response 1: Thank you for your feedback. The “stroke and its associated complications” sentence have been added to the manuscript for more clarification. (Page 1, line 16 & 44)

Comments 2: Spasticity cannot be called a problem, it is rather a limitation of full activity.

Response 2: We acknowledge and agree that spasticity, characterized by increased muscle tone and stiffness, can restrict or impede the full range of motion and functional abilities of individuals. Upon a careful review of the manuscript, we have ensured that the term "spasticity" is used in a manner that aligns with this understanding.

Comments 3: Exercises and electrostimulation may not be crucial.

Response 3: Thank you for your feedback. Based on our research and the existing body of literature, exercises and electrostimulation have been widely recognized as valuable interventions for various conditions. They have demonstrated effectiveness in improving muscle strength, enhancing functional abilities, promoting rehabilitation, and aiding in the recovery process. In our study, we specifically focused on the benefits of exercises and electrostimulation in addressing the targeted outcomes (e.g. plantar flexor muscle spasticity). Our findings support the notion that these interventions play a crucial role in improving the condition under investigation.

Comments 4: Walking efficiency was not measured, but rather quality (subjective assessment) or quantity (objective assessment, e.g. 3D gait analysis)

Response 4: Thank you for your comment. We appreciate your observation regarding the measurement of walking efficiency in our study. It is important to note that the primary aim of our study was to evaluate the effect of NMES on spasticity rather than conducting a comprehensive analysis of walking efficiency.

While we agree that measuring walking efficiency using more objective assessments, such as 3D gait analysis, would provide further clarification, it was beyond the scope of our study to delve into that aspect. Our focus was specifically on assessing the impact of NMES on spasticity. Your suggestion of incorporating walking efficiency measurements in future studies is valuable.

Comments 5: There is no information about who conducted the MAS study or what the group selection process was (random, random, or coin tossing). There is no information about the state of spasticity, how many years after the stroke, how long the exercises lasted and the type of exercises - standard or neurodevelopmental

Response 5:

  • On page 5&6, lines 148-152 there is information about the MAS.
  • On page 5, lines 137-141: The allocation assignment for the study was randomly assigned to either the NMES active or NMES sham group in a 1:1 ratio. This randomization was performed using an online randomization website, which can be accessed at the following URL: https://www.graphpad.com/quickcalcs/randomize1.cfm.
  • On page 8: Table 3 provides information about the time since the stroke, and it is mentioned.
  • On page 5&7: The type of exercise is described in detail in table 1.

Reviewer 2 Report

Comments and Suggestions for Authors

Thank you for the opportunity to review the manuscript. The manuscript presents a clinical trial focused on the impact of neuromuscular electrical stimulation on spasticity and walking performance in the context of chronic stroke. Please find my suggestions.

Objectives:

The authors wrote:

Lines 18-20: “Therefore, this study aimed to evaluate the effects of neuromuscular electrical stimulation (NMES) on spasticity and walking performance among individuals with chronic stroke.”

Lines 80-82: “Therefore, this study aimed to contribute to the growing body of evidence and facilitate the development of NMES to optimize spasticity management and enhance the rehabilitation outcomes of stroke patients.”

Lines 242-243: “This study aimed to evaluate the effectiveness of using NMES a combined with CRP on plantar flexor muscles spasticity in stoke patients.”

I suggest always describing the same objectives.

Abstract:

-Please, describe the date of the study.

-Line 27: “…Ashowrth Scale (MAS), gait speed measured by 10-meter walk test…”. I suggest writing the acronym (MWT).

Introduction:

Lines 68-73: reference 14 (year 2000): Is this systematic review the most recent?

Material and Methods:

-Please, describe the date of the study.

-Line 119: Please, describe “FES”.

-Table 1: Please, describe “COG”

-Lines 180-187: Considering that sham stimulation is different from active stimulation, how was the therapist considered “blind” in this study?

Discussion

-Line 243: Please, correct “stroke”.

-The discussion was developed using, for example, references 15, 22, 23, 25 and 26 from 2013, 2004, 2014,1997 and 2018, respectively. The authors could include more recent references.

Limitation

Another limitation could be “the study was not multicenter”.

References

Please, review references, for example, 3, 12 and 15. Are they from articles or books?  Which pages?

Author Response

Response to Reviewer 2

Objectives:

The authors wrote:

Lines 18-20: “Therefore, this study aimed to evaluate the effects of neuromuscular electrical stimulation (NMES) on spasticity and walking performance among individuals with chronic stroke.”

Lines 80-82: “Therefore, this study aimed to contribute to the growing body of evidence and facilitate the development of NMES to optimize spasticity management and enhance the rehabilitation outcomes of stroke patients.”

Lines 242-243: “This study aimed to evaluate the effectiveness of using NMES a combined with CRP on plantar flexor muscles spasticity in stoke patients.”

Comments 1: I suggest always describing the same objectives.

Response 1: Thank you for your suggestion. The objectives of the study have been revised accordingly in the manuscript. “this study aimed to evaluate the effects of neuromuscular electrical stimulation (NMES) combined with conventional rehabilitation program (CRP) on plantar flexor muscle spasticity and walking performance among individuals with chronic stroke”

Abstract:

Comments 2: Please, describe the date of the study.

Comments 3: Line 27: “…Ashowrth Scale (MAS), gait speed measured by 10-meter walk test…”. I suggest writing the acronym (MWT).

 Response 2&3: The suggestions have been incorporated into the manuscript.

Introduction:

Comments 4: Lines 68-73: reference 14 (year 2000): Is this systematic review the most recent?

Response 4: The reference has been changed.

Material and Methods:

Comments 5: Please, describe the date of the study.

Comments 6: Line 119: Please, describe “FES”.

Response 5&6: the date has added and the FES is described in the manuscript.

Comments 7: Table 1: Please, describe “COG”

Response 7: COG is described in the manuscript.

Comments 8: Lines 180-187: Considering that sham stimulation is different from active stimulation, how was the therapist considered “blind” in this study?

Response 9: In this study, the therapist was aware of the allocation of participants, while the assessor remained blinded to the allocation of participants.

Discussion

Comments 9: Line 243: Please, correct “stroke”.

Response 9: The aimed of the study has been revised with correct spelling of “stroke”

Comments 10: The discussion was developed using, for example, references 15, 22, 23, 25 and 26 from 2013, 2004, 2014,1997 and 2018, respectively. The authors could include more recent references.

 Response 10: Thank you for your suggestion. Three more references have been added to the discussion. (Page 9, line 277-284)

Limitation

Comment 10: Another limitation could be “the study was not multicenter”.

Response 10: Another limitation has been added to the manuscript. (Page 10, line 323 -326)

References

Comment 11: Please, review references, for example, 3, 12 and 15. Are they from articles or books?  Which pages?

Response 11: The references have been reviewed carefully.

Reviewer 3 Report

Comments and Suggestions for Authors

The aim of this manuscript was to evaluate  the effects of neuromuscular electrical stimulation (NMES) on spasticity and walking performance among individuals with chronic stroke. These are my comments and suggestions:

Abstract:

Please, check your abstract for typographical errors (Ashowrth Scale).

Introduction:

Please, explain what would be the physiological background of NMES in stroke patients. What would be the mechanism of working?

Methods:

Please, explain how did you decide that your sample size is adequate. Figure 1 has low quality, it should be improved.

Results:

I advise calculating effects size since your sample was quite small.

Discussion:

Please comment is the NMES really the best evidence-based therapy for spasticity after stroke compared to other approaches?

Author Response

Response to Reviewer 3

Abstract:

Comments 1: Please, check your abstract for typographical errors (Ashowrth Scale).

Response 1: Thank you for bringing this typo to our attention. The word has been corrected.

Introduction:

Comments 2: Please, explain what would be the physiological background of NMES in stroke patients. What would be the mechanism of working?

Response 2: An explanation has been added to the manuscript (page 2, line 67-71)

Methods:

Comments 3: Please, explain how did you decide that your sample size is adequate. Figure 1 has low quality, it should be improved.

Response 3: The sample size was a limitation of the study and it’s acknowledge in the limitation paragraph. I agree that the figure has poor quality and we replaced it with a high quality one.

Results:

Comments 4: I advise calculating effects size since your sample was quite small.

Response 4: Thank you for your valuable comment and suggestion. We totally agree with you that calculating the effect size would be helpful for readers and clinicians.  However, the main outcome measure (spasticity using the Ashworth Scale) was not truly a ratio scale (continuous variable), and it is likely to be an ordinal scale with very limited categories. Therefore, we did not calculate the effect size for the aforementioned reasons, in addition to the normality violation for ordinal scales such as the Ashworth scale.

Discussion:

Comment 5: Please comment is the NMES really the best evidence-based therapy for spasticity after stroke compared to other approaches?

Response 5: NMES has been used as a therapy for spasticity and has shown promising results in some studies. It is important to note that the effectiveness of any treatment can vary depending on several factors. The aligns with this statement has been implemented on page 10, lines 311–320.

Round 2

Reviewer 3 Report

Comments and Suggestions for Authors

I am happy with most of the improvements. However, I am worried regarding the small sample size.